# Identifying and Comparing Easily Accessible Frameworks for Assessing Soil Organic Matter Functioning

Lúcia Barão [1,*], Abdallah Alaoui [2]  and Rudi Hessel [3]

1   Center for Ecology, Evolution, and Environmental Changes (cE3c) & CHANGE-Global Change and Sustainability Institute, Faculdade de Ciências da Universidade de Lisboa, Campo Grande, 1749-016 Lisbon, Portugal
2   Institute of Geography, University of Bern, Hallerstrasse 12, 3012 Bern, Switzerland
3   Wageningen Environmental Research, Droevendaalsesteeg 3, 6708 PB Wageningen, The Netherlands
*   Correspondence: albarao@fc.ul.pt

**Abstract:** Soil organic matter (SOM) stocks are crucial for soil fertility and food provision and also contribute to climate change adaptation and mitigation. However, assessing SOM changes in cropping systems is difficult due to the varying quantity and quality of input data. SOM processes have been described by several models, but these are complex and require high amounts of input data. In this work, we identified and selected frameworks that simulate SOM pools and stocks as well as the effects of different management practices. We also required that the frameworks be easily accessible for farm-related end users and require limited and accessible amounts of input data. In all, six frameworks met our inclusion criteria: SOCRATES (Soil Organic Carbon Reserves and Transformations in EcoSystems), CCB (CANDY and-Carbon Balance), AMG, CENTURY, CQESTR, and RothC (Rothamsted Carbon Model). We collected information on these frameworks and compared them in terms of their accessibility, the model time steps used, the nutrient cycles included in the simulation, the number of SOM pools, and the agricultural management options included. Our results showed that CCB was the most robust of the frameworks considered, while AMG, CQESTR, and RothC performed the least well. However, all frameworks have strengths which may match the specific requirements and abilities of individual users.

**Keywords:** sustainability; soil; farming; organic pools; tools

## 1. Introduction

Recent initiatives, such as the United Nations declaring 2015 as the International Year of Soils, and the French "4 per 1000" initiative, illustrate a worldwide desire to increase soil organic matter stocks for soil fertility and food security purposes, as well as for climate change adaptation and mitigation. Soil organic matter (SOM) plays a central role in soil fertility and soil water retention [1], and thus contributes to agroecosystem productivity and to food security. SOM also enables agroecosystems to adapt to a changing climate with less frequent, less regular, and more extreme rainfall, as well as associated erosion problems [2–4]. The quantitative data of soil organic carbon (SOC) pools [5] and fluxes [6] show that small changes in world SOC stocks could either aggravate or mitigate global greenhouse gas (GHG) emissions, an idea suggested by Balesdent et al. (1999) [7], and further developed in recent articles [8–12].

Maintaining the long-term productivity of soil is an important goal in sustainable agriculture [13]. SOM is a key factor in arable crop production systems [14]. For this reason, a knowledge of its structure, functions and dynamics helps to increase soil quality [15]. However, assessing SOM changes in cropping systems (CSs) is challenging. SOM content in soil changes slowly because added organic matter is subject to various processes, so that the SOM pools that are built up in the soil are quite different to the organic matter that is added in terms of both quantity and content. In addition, SOM encompasses all organically

based compounds that can be found in the soil matrix. These diverse chemical and physical characteristics, which also involve living organisms such as microbes, make the SOM pool one of the most complex entities to measure and/or simulate. When assessed in the laboratory, different SOM pools are often individualized by their affinity to extract products or differentiated on the basis of methods that separate pools with different physical properties. Chemical methods isolate specific fractions such as microbial biomass, carbohydrates [16], polysaccharides [17,18], and fluvic/humic acids [19] or more encompassing fractions such as active organic carbon [19,20], hot-water extractable organic C [21,22], biologically active C [23], labile organic matter [24,25], chemically stabilized organic C [18], and recalcitrant organic matter [16,19]. Some of these fractions are used to evaluate short-term consequences related to land-use changes and different agricultural/soil management practices (including conservation agriculture) while also serving as soil quality indicators [26,27]. Physical methods separate SOM fractions based on size and/or density, suggesting that different aggregates are impacted differently. These methods often separate particulate organic matter (POM)—which is considered very responsive to the quantity and quality of crop residues added to the soil [17,21,28]—from minerals associated organic carbon (MAOC) [21] which are also affected by the addition of crop residues [28] and, in some cases, by tillage practices [18].

At the same time, several models and approaches have been developed for describing soil processes and simulating SOM pools. These include mathematical representations of physical, chemical, and biological processes, requiring large and complicated input datasets [29]. SOM pools in models are often individualized according to different decomposition rates, but the link between simulated SOM pools and actual laboratory measurements remains weak, thus limiting model performance. In short, models can simulate with high accuracy the concentration and dynamics of different pools, such as SOM, over long time periods. However, these are not only time consuming, but also require input data that are not easily collected or measured.

Farmers and other farm-related stakeholders often face difficulties in using these frameworks to obtain practical advice and to determine the management practices they should adopt to maintain and improve SOM in their soils. In this regard, simpler frameworks may be useful, even if their level of accuracy is lower. Many such frameworks have been suggested that differ in terms of processes simulated, means of input data collection, output availability, etc. The objectives of this study are twofold; first, to select and characterize frameworks that have been used to assess SOM changes at the farm level and which are user friendly for farmers; second, to compare these frameworks and establish appropriate conditions for improved use.

## 2. Materials and Methods

### 2.1. Framework Selection Criteria

To achieve the first goal of this work, we established 5 criteria to narrow our choice of frameworks simulating SOM pools and stocks. Using these criteria, we sought the following:

1. Frameworks that simulate SOM concentrations and stocks in different cropping systems with different management practices and options. It is important that the framework is able to simulate SOM for different crops managed in different ways by individual farmers. The management choices made by farmers critically affect the accumulation of SOM in soils. The inclusion of such choices in the framework is therefore essential if useful guidance for farmers is to be obtained.
2. Frameworks that simulate a maximum of 3 SOM pools. It is important not to operate with frameworks that consider more than 3 pools. The separation of SOM pools in models and in nature involves different processes, and any link between simulated and measured pools is difficult to obtain.
3. Frameworks that are applicable at the farm level and are able to provide local answers to farmers under specific local conditions.

4. Frameworks that include a platform already accessible for users, either online or through download. This ensures that the framework is readily available and user friendly for farmers.
5. Frameworks requiring low amounts of easily accessible data for input. This ensures that the framework is not an overcomplicated model which might be very accurate but requires too much input data to be of practical benefit.

To identify suitable frameworks simulating SOM pools, we searched published articles related to SOM pool assessment and measurement. This search was first carried out among review papers and then, among relevant articles cited in the review papers. Only frameworks fulfilling the above criteria were considered. These were then described in terms of general information such as authors, download links, type of product, model time step, cycles included, number of SOM pools, soil management options included, and the overall model scheme. We also set out the input data required for each framework, covering such essential variables as soil properties and climatic information, as well as details of crops and crop management.

*2.2. Framework Comparison*

The second goal of our study was to compare the selected frameworks and highlight those which were most robust, i.e., those which showed high scores in the majority of compared characteristics. Firstly, we described the different selected frameworks in terms of the following characteristics: (1) Type of product—how accessible the product is for users, i.e., whether it is an online tool, or downloadable software; (2) model time step—whether the model uses a daily, weekly, monthly or annual step; (3) cycles included—whether carbon, nitrogen, phosphorus, etc., are simulated in the model; (4) SOM pools—the number and type of SOM pools simulated; and (5) soil management practices simulated—the options that the model is able to simulate. It is worth noting here that a framework might have advantages, such as being a simple online product accessible to users without a scientific background, and limitations, such as a low number of cycles and/or SOM pools simulated. Secondly, we sought to provide an overview of the average performance of each framework. For this purpose, we scored each framework on the relevant characteristics described above (type of product, model time step, cycles included, SOM pools and soil management practices simulated) with values between 1 and the highest number for each characteristic (ranging from the least to the most complete performance in that characteristic). This evaluation provided useful framework performance information for each individual characteristic and formed the basis of an overall score for all integrated characteristics.

## 3. Frameworks Assessing SOM

In total, six frameworks satisfied our selection criteria, namely: Soil Organic Carbon Reserves and Transformations in EcoSystems (SOCRATES), CANDY and Carbon Balance (CCB), AMG, CENTURY, CQESTR, and RothC. The main characteristics of each of these frameworks are described below. Table 1 presents the required input for each framework (required soil properties, climatic information, and crop/management details of the field). Table 2 shows general and systematic information including the link to each framework, the type of product provided, the model time step used, the cycles included in the modeling, and the number of SOM pools included, as well as the soil management practices simulated.

**Table 1.** Input data required for all 6 frameworks, categorized by soil properties, climatic information, and details of crops and/or crop management.

| Framework | Input Data | | |
|---|---|---|---|
| | **Soil Properties** | **Climatic Information** | **Crop and/or Management Details** |
| SOCRATES | Soil clay content or CEC, bulk density, initial soil organic C | Average annual precipitation, mean annual temperature | Crop yield, crop types |
| CCB | Bulk and substrate density, soil moisture at field capacity and wilting point, texture indicator, saturated conductivity | Air temperature, global radiation, precipitation | Crop rotation and average yields before initial point |
| AMG | Clay and carbonate contents, pH, SOC content, C:N ratio, bulk density of soil and depth | Mean annual temperature, precipitation, evapotranspiration | Crop yields and crop residues management, soil tillage (depth), irrigation, EOM application (type, rate, and date) |
| CENTURY | Soil texture, initial soil C, N, P, and S levels, soil N inputs | Atmospheric N inputs, air temperature, precipitation | Lignin content of plant material, plant N, P, and S content |
| CQESTR | Number and depth of layers, initial SOC, recent SOC, soil bulk density, soil texture, soil drainage | Temperature, precipitation | Above- and below-ground biomass additions, N content of residues and amendments |
| RothC | Depth of soil layers, clay content, soil cover | Air temperature, precipitation, pan evaporation | Plant residues and/or farmyard manure, decomposability of the incoming plant material: DPM/RPM [1] ratio |

[1] Decomposable plant material (DPM) and resistant plant material (RPM).

### 3.1. Soil Organic Carbon Reserves and Transformations in EcoSystems (SOCRATES)

SOCRATES is a process-based simulation model developed in Australia, designed to estimate changes in topsoil organic carbon (SOC) using a minimal dataset set of soil, climate, and biological inputs [30,31]. The software is available online and simulates the carbon cycle with three SOM pools: two microbial pools (unprotected and protected pools) and the humus fraction. The parameters used in the simulation are already fixed for certain conditions based on previous studies and/or are calculated from input data such as soil temperature and soil moisture. The input data required include the following: soil properties (i.e., soil clay content or CEC, bulk density, and initial SOC); climatic data such as average annual precipitation and mean annual temperature; and, finally, crop type and crop yield obtained (if available). The model also allows the simulation of tillage management interference with the SOM pools. SOCRATES was successful in predicting SOC changes in eighteen long-term studies of crops, pasture, and forestry in North America, Europe, and Australasia. These trials ranged from 8 to 86 years in duration and covered a wide range of climates and soil types [30].

### 3.2. CANDY Carbon Balance

The CANDY Carbon Balance (CCB) model was developed in Germany to give farmers a tool to calculate short-term dynamics of nitrogen transformations and long-term changes in soil carbon content [32–34]. It is a software product with a detailed description that simulates the carbon and nitrogen cycles and includes three SOM pools based on their decomposition potential: an active pool, where the mineralization takes place; a stabilized pool, representing the passive but decomposable part of the SOM; and a long-term stabilized pool that is regarded as inert. The process modeling parameters have previously been set for different conditions, including a wide range of sources of fresh organic matter (FOM), and this has been successfully validated in some studies [35]. The input data required to run the model include the following: soil properties (bulk and substrate density, soil moisture at field capacity and wilting point, texture indicator, and saturated conductivity); climatic information (air temperature, global radiation, and precipitation); and, finally, crop characteristics (crop rotation details and average yields.) The CANDY Carbon Balance

(CCB) model has been validated using a dataset from 40 long-term experiments carried out in Central Europe, including 391 treatments with a total of 4794 $C_{org}$ observations [32].

### 3.3. AMG

This framework was developed in France and represents another example of an Excel tool which simulates the evolution of the carbon content of soils on a field scale for a given cropping system [36]. AMG is a decision support tool: the farmer can see the long-term effects of alternative practices, compared with those he/she currently applies. It simulates the carbon cycle through only two SOM pools, one active and the other stable. The parameters required for the process modeling are dependent on, and calculated from, input data (soil, plant, and climate) provided by the users. Input data for parameterization and initial conditions include soil properties (clay and carbonate contents, pH, SOC content, bulk density, and carbon/nitrogen ratio), climatic variables (mean annual temperature, precipitation, and evapotranspiration), crop information (crop yield and crop residue management), and management practice options (soil tillage depth, irrigation periods, and exogenous organic matter (EOM) application (type, rate, and date)). An examination of costs, physical constraints, or forms of work organization associated with various technical options can also be included, and this can modify recommendations provided by the framework, adapting them either to the context of operation, or to the means and objectives of the farmer. This tool has already proved effective for SOC prediction in long-term field experiments with EOM application [36] and for various possibilities of crop rotations [37].

### 3.4. CENTURY

CENTURY Agroecosystem Version 4.0 is a software product developed in the USA to deal with a wide range of cropping system rotations and tillage practices. It is used for system analysis of the effects of management and global change on productivity and sustainability of agroecosystems [38,39]. It simulates carbon, nitrogen, phosphorus and sulfur cycles and also tackles organic matter decomposition processes by simulating three SOM pools: active, slow, and passive fractions. Parameterization is dependent on data available in the literature and on input data concerning soil, plant, and climate conditions. To successfully run the model, users need to know soil texture and initial soil carbon (C) nitrogen (N), phosphorus (P) and sulfur (S) concentrations, as well as atmospheric and soil N inputs. The lignin content of plant material, as well as plant N, P, and S content, is also required. Essential climatic variables include air temperature and precipitation. The latest release of the model is able to simulate complex agricultural management systems including crop rotations, tillage practices, fertilization, irrigation, grazing, and harvest methods. Long-term experimental studies have shown that CENTURY can simulate different fertilization effects on SOC dynamics under variable climate and soil conditions [39].

### 3.5. CQESTR

The C model CQESTR, pronounced 'sequester', was developed in the USA to evaluate the effect of agricultural management practices on short- and long-term SOM dynamics [40]. It is available as a software product which simulates the carbon cycle using only one pool of SOM that is modeled in a continuum. The information required to run the model includes the number and depth of the soil layers considered by the user, and the soil properties of these layers, i.e., initial SOC, bulk density, soil texture, and soil drainage. Required climatic factors include temperature and precipitation, as well as boundary conditions, above- and below-ground biomass additions, and the N content of residues and amendments. Management practices included in the model are tillage and crop rotation. The model was calibrated with results of a long-term experiment using soil carbon data and then validated at 11 independent sites, showing that simulated and measured values were very close [41].

**Table 2.** Frameworks characterization showing its authors affiliation, type of product, model time step, cycles simulated, number of SOM pools included, type of management simulated, and the model scheme.

| Characteristics | Soil Organic Carbon Reserves and Transformations in EcoSystems (SOCRATES) | CANDY and Carbon Balance (CCB) | AMG | CENTURY | CQESTR | RothC |
|---|---|---|---|---|---|---|
| Affiliation of authors | School of Natural Resource Sciences, Queensland University of Technology | Helmholtz–UFZ | INRA | Natural Resource Ecology Laboratory, Colorado State University | USDA-ARS Columbia Plateau Conservation Research Center, Pendleton, Oregon | Rothamsted Research |
| Link | http://socrates.n2o.net.au/main accessed on 26 December 2022 | https://www.ufz.de/index.php?de=39724 accessed on 26 December 2022 | http://www.simeos-amg.org/ accessed on 26 December 2022 | https://www.nrel.colostate.edu/projects/century/century-obtain.php accessed on 26 December 2022 | https://www.ars.usda.gov/pacific-west-area/pendleton/columbia-plateau-conservation-research-center/docs/cqestr/ accessed on 26 December 2022 | https://www.rothamsted.ac.uk/rothamsted-carbon-model-rothc accessed on 26 December 2022 |
| Type of product | Online software | Software with detailed description available | Microsoft Excel | Software | Software | Software |
| Model time step | Weekly | Daily | Annually | Monthly | Daily | Monthly |
| Cycles included | Carbon | Carbon and Nitrogen | Carbon | Carbon, Nitrogen, Phosphorus, Sulfur | Carbon | Carbon |
| SOM pools | 3 pools | 3 pools | 2 pools | 3 pools | 1 pool | 3 pools |
| Soil management | Tillage | Soil tillage, irrigation, application of mineral fertilizer, application of OM, sowing (emergence), and harvest methods | Tillage, irrigation, input of organic fertilizer | Fire, harvest methods, grazing and cultivation | Tillage, residue cover | Input of farmyard manure and plant residues |

**Table 2.** *Cont.*

| Characteristics | Soil Organic Carbon Reserves and Transformations in EcoSystems (SOCRATES) | CANDY and Carbon Balance (CCB) | AMG | CENTURY | CQESTR | RothC |
|---|---|---|---|---|---|---|
| Model scheme | Reprinted from [30] | Reprinted from [33] | Reprinted from [36] | Reprinted from [38] | Reprinted from https://www.ars.usda.gov/pacific-west-area/pendleton/columbia-plateau-conservation-research-center/docs/cqestr/ accessed on 26 December 2022 | Reprinted from [42] |

*3.6. RothC*

This model was developed in the UK to simulate organic carbon in non-waterlogged topsoil dependent on the influence of factors such as soil type, temperature, soil moisture and plant cover on the turnover process. It is available as a software product for download and simulates the carbon cycle with three pools for SOM: microbial biomass, humified organic matter, and inert organic matter. Additionally, there is an input of fresh organic matter through residues and/or manure [42]. The input data includes information on the depth of layers simulated as well as their clay content and soil cover. Required climatic information includes temperature, precipitation and pan evaporation. Boundary conditions require information on the input material added to the soil, in particular its decomposability, expressed as the ratio of decomposable plant material (DPM) and resistant plant material (DPM/RPM) [43]. Aguilera et al. [44] presented details of SOC balance in an area of cultivated land in Spain from 1900 to 2008, as a model of Mediterranean-type industrialized agriculture. They combined RothC model parameters with humification coefficients using the humified soil organic carbon model (HSOC) to simulate changes in SOC stocks. RothC was also combined with two empirical approaches for quantifying carbon inputs from above- and below-ground crop residues and tested on 439 SOC data series from 36 long-term arable field experiments in Central and Northern Europe. Although the validation showed good correlation between simulated and measured values, the results also showed that an overestimation of C sequestration from above-ground crop residues [45].

## 4. Comparison of the Frameworks

*4.1. Framework Characteristics*

The six frameworks described in this work were selected because of their availability, ease of use, and ability to present a complex environment such as soil in simple terms. However, and as summarized in Table 2, the six compared frameworks all exhibit differences in five important characteristics, which we now set out in detail.

### 4.1.1. Type of Product

The most easily accessible framework is SOCRATES, because any user (with or without modeling knowledge) can access the online page and provide the required information. The information required to run the model is minimal, which makes it convenient for occasional users. AMG is provided as an Excel tool, which is also convenient for those individuals without a technical modeling background. These include many farmers who seek information on SOM evolution in their fields. Finally, all the other frameworks are accessible through download or as software products, and this involves a higher level of difficulty for users without a technical background (Table 2). However, these frameworks are still simpler and require fewer input data than other more sophisticated models which are available, such as SWAT (Soil and Water Assessment Tool) [46], RZWQM (Root Zone Water Quality Model) [47], and MohidLand [48].

### 4.1.2. Model Time Step

Both CCB and CQUEST have a daily time step, which means that the model calculates pools and flows for each day. This enables the consideration of daily variations in weather (such as temperature and rain) which can be substantial. By such means, the quality of results can be improved [49]. The SOCRATES framework provides a weekly time step. However, the user does not need to enter data on a weekly basis. Instead, the framework works internally using available weather data from the location provided by the user. CENTURY and ROTHC both operate with monthly time steps, while AMG uses an annual time step, representing a gradient of precision loss in calculating the SOM outputs in a specific field. Although smaller time steps may, in general, indicate that the model has a higher output precision, it may also imply that some input data must be provided with more detail, as is the case for climatic information (Table 1).

### 4.1.3. Cycles Included

The carbon cycle is closely interconnected with the cycles of nitrogen, phosphorus, sulfur, and other nutrient elements. The concentration and specification of pools from other nutrients also affects the concentration of SOM in different soil layers [50]. For this reason, integrating non-carbon cycles into the model leads to higher precision in modeling the processes that govern SOM concentration, such as organic matter decomposition and microbial immobilization [51]. Frameworks that can simulate more than just the carbon cycle will produce higher precision in SOM simulation, because they are able to capture more interactions [52]. However, the inclusion of such cycles requires a higher number of input data and/or data from the literature related to the newly included pools. For this reason, we consider CENTURY to be the most complete framework, because it also includes nitrogen, phosphorus and sulfur. In this regard, CCB can be considered an intermediate framework, because it includes only nitrogen in addition to carbon. The remaining frameworks include the carbon cycle only, and thus maintain a simpler approach.

### 4.1.4. SOM Pools

Complex models can easily include many SOM pools in their simulation. The underlying idea is to capture the complexity of the organic matter decomposition gradient that exists in soils [53]. On the one hand, easily decomposable SOM has a short lifetime, is normally more abundant in the upper layers, and often forms complexes with other compounds. On the other hand, recalcitrant organic matter is more frequently found in subsoil, and can endure for a longer period [54]. Between these two extremes, there is a continuum of other SOM pools, each with its own specific decomposition rate. Due to the difficulty of simulating all of them, modelers often break them into a discrete gradient with variable decomposition rates [55]. However, increasing the number of SOM pools also increases the requirement for input data that are not easily available, such as the initial concentration of each of these pools both within soil layers and along the layers [56]. All frameworks analyzed here include a maximum of three SOM pools to decrease complexity. CENTURY, RothC, CCB and SOCRATES all include three pools, with low, medium and high decomposition rates, and therefore capture the degradation gradient. AMG includes only two pools: one highly degradable, and the other more recalcitrant. Finally, CQETR includes a single pool only.

### 4.1.5. Soil Management Practices Simulated

The inclusion of management options is a positive attribute of any framework. Farmers, stakeholders and other end users are typically interested in forecasting the effects of agricultural management practices which they may choose to implement concerning the amount of SOM in farm fields. Any method used to achieve this should not only calculate levels of organic matter under current conditions but should also predict its future concentrations under various hypothetical conditions. CCB simulates five different management choices concerning tillage, fertilization, irrigation, cultivation, and harvest methods. CENTURY includes four management options, and AMG includes three. CQESTR and RothC both include two options concerning tillage and the input of organic manure and plant residues. Finally, SOCRATES simulates only tillage, and its range is therefore more limited compared to the others.

It is also of interest to analyze the different soil management options simulated by the different frameworks. The prediction of operations likely to improve crop yield is a valuable asset in the assessment of economic sustainability [57]. The simulation of fertilization [58] and irrigation [59] effects is especially important, because of issues related to the changing price and availability of fertilizers and water. The ability to simulate these effects constitutes the most important advantage of the CCB, AMG and RothC frameworks. However, simulation of other operations such as tillage [60], residue cover [61] and the input of organic fertilizer and/or manure [62] can be also appealing for farmers.

### 4.2. Framework Scoring Evaluation

The previous comparison showed that each framework has its own input data requirements (Table 1), strengths, and weaknesses with respect to the particular skills and goals of users (Table 2).

In general, all the frameworks considered are applicable and useful. If a farmer wishes to simulate the SOM concentrations in his/her land and does not have any background in information science, then the SOCRATES tool is the best option. However, if a specific agricultural stakeholder is interested in the SOM effect on other nutrient cycles (especially concerning fertilizer dose application) then CENTURY is the best option, though it may lack some precision in its assessment of other characteristics (Table 2). When comparing the scores achieved by each framework for each characteristic (Figure 1), we find that the CCB framework attains the highest accumulated total, and is thus best able to fulfill all characteristics, although with a lower score on the product accessibility. It is closely followed by CENTURY and SOCRATES, which achieved similar scores (13 and 11, respectively) but which differ in terms of their strengths and weaknesses. While SOCRATES has very good product accessibility and is able to simulate several SOM pools, CENTURY is strong in terms of the number of cycles included in the simulation, as well as the number and variety of agricultural management options simulated. Finally, we find equally low scores for AMG, CQUEST, and ROTHC. While AMG achieves generally low and medium scores across all characteristics, CQUEST is very strong in terms of its time steps but weak in the remaining characteristics. RothC also scores highly in terms of the SOM pools simulated but less well in respect of the other characteristics (Figure 1).

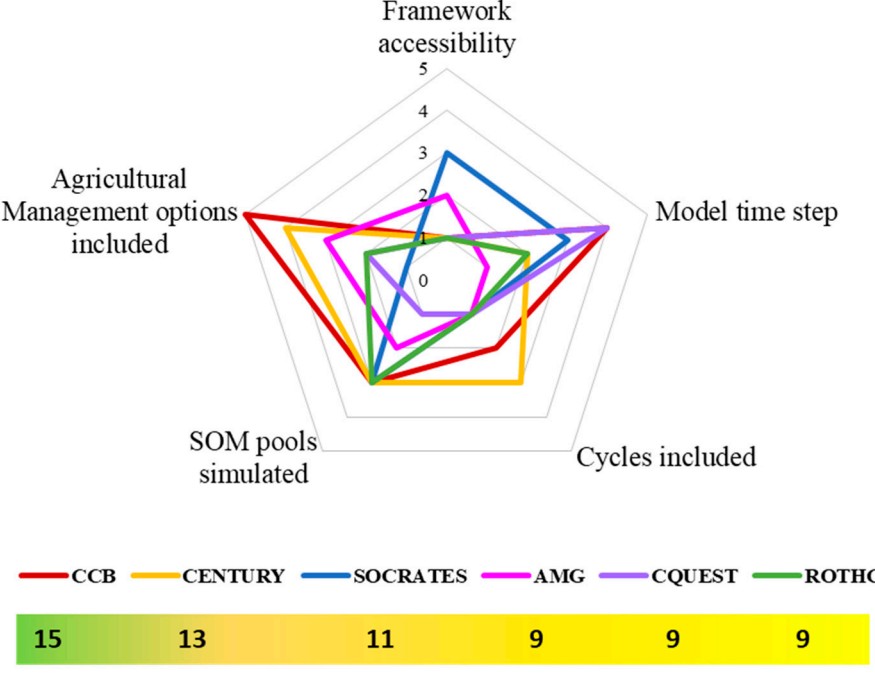

**Figure 1.** Scores obtained for each characteristic from each framework. The summed scores are presented below, from green (highest) to yellow (lowest).

## 5. Conclusions

In this study, six simple and user-friendly frameworks simulating SOM pools in different cropping systems were selected, characterized, and compared: SOCRATES, CCB, AMG, CENTURY, CQRESR, and RothC.

All frameworks analyzed here are available in an online platform, simulate a maximum of three SOM pools, are applicable at the farm level, include the simulation of different management practices, and require a low amount of input data to operate. Despite these common attributes, they nonetheless differ in many other characteristics, which are

representative of the framework philosophy and goals. These characteristics include the number of SOM pools simulated, the time step used, the nutrient cycles included in the modeling and the type of soil management practice involved.

When comparing the frameworks based on these characteristics, it becomes clear that, depending on the user's needs and skills, each one might represent the best choice for individual farmers or related stakeholders. However, when analyzing the average performance of the six frameworks, we conclude that the CCB framework is the most robust, with high scores in all characteristics except for product accessibility, while AMG, CQRSTR and RothC all score poorly in several characteristics, indicating weakness in these areas.

**Author Contributions:** Conceptualization: L.B., A.A., R.H.; methodology: A.A., L.B.; investigation: L.B., A.A.; writing—original draft preparation: L.B., A.A., R.H. All authors have read and agreed to the published version of the manuscript.

**Funding:** This research received funding from the European Union's Horizon 2020 research and innovation program under grant agreement No. 677407 (SoilCare project). A.A. also received funding from the European Union's Horizon 2020 program for research & innovation under grant agreement No. 862568 (SPRINT project).

**Data Availability Statement:** Not applicable.

**Conflicts of Interest:** The authors declare no conflict of interest.

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
