# Peer review of "Identifying and Comparing Easily Accessible Frameworks for Assessing Soil Organic Matter Functioning"

_agronomy, doi:10.3390/agronomy13010109_

Round 1

Reviewer 1 Report

I think this paper has merit as an interesting comparison of user-friendly models. However, it needs extensive copy-editing and a little restructuring.

The introduction needs a thorough revision. The language is not always correct. More importantly, the structure is unclear, as the deep information on SOM pools doesn’t support the comparison among models. The clarity of the language improved considerably starting in Section 3, and the models are clearly described. But several points of confusion remained, highlighted below.

The paper would benefit from a clear Methods section, describing the selection and assessment process in more detail. Lines 107-116 get a good start on this, and L 340-346 gives some sense of the assessment, but these points should be separated and expanded.

L 20 it’s not clear why only some of the acronyms are spelled out.

L 23 “both” shouldn’t refer to 3 models

L 45 not very clear- does “input data” refer to models?

L 54 SOM is not really a pool of C

L 66-71 very mixed up language. Also it’s not clear why SOM pools should follow from the objectives stated in the previous paragraph.

L 312-315 do the different management ‘types’ or ‘options’ refer to aspects of management that can be controlled by the models? Or are they, in the case of CCB, 5 pre-determined ‘types’ representing 5 different complete management styles?

L 318 What do you mean by ‘separate’? You mean the user can manipulate different aspects independently?

L 324 what do you mean by ‘high value’? That they are important for stakeholders or important for SOM estimates?

L 339-341 From “The same is applicable”… to “beneficial” is not needed.

Conclusion: needs a thorough grammar review for clarity.

Author Response

C1: I think this paper has merit as an interesting comparison of user-friendly models. However, it needs extensive copy-editing and a little restructuring. The introduction needs a thorough revision. The language is not always correct. More importantly, the structure is unclear, as the deep information on SOM pools doesn’t support the comparison among models. The clarity of the language improved considerably starting in Section 3, and the models are clearly described. But several points of confusion remained, highlighted below.

R1: Thank you so much for your general comment on the paper. We followed your advice and we made several modifications on the article structure to clarify the work performed here: 1) We have removed the section about SOM pools description. Instead, we have moved some of this text into the Introduction and alter Introduction itself as suggested.  The idea is to show first that SOM is important for agriculture sustainability and food provision, then show that SOM pools are measured in laboratory in many ways and linked to soil functions and soil quality assessment and finally that models and other frameworks also describe SOM pools in a different way and the link between these two is difficult, which makes models with many SOM pools simulated highly complex; 2) We have added a section “Materials and Methods” where we described how we made the framework selection and the criteria used and also how we made the comparison between frameworks and the characteristics used for this comparison. The text from section 3 and 4 where this information was written previously, was removed; 3) Section 4.2 was renamed “Frameworks scoring evaluation” and the summary became a reflection on the different scoring achieved by each framework.

C2: The paper would benefit from a clear Methods section, describing the selection and assessment process in more detail. Lines 107-116 get a good start on this, and L 340-346 gives some sense of the assessment, but these points should be separated and expanded.

R2: We took this comment into consideration, and we added a Materials & Methods section as described above.

C3: L 20 it’s not clear why only some of the acronyms are spelled out.

R3: Thank you for this comment. We understand the confusion, but the reason is that some of the acronyms are in fact the frameworks name and cannot be spelled out.

C4: L 23 “both” shouldn’t refer to 3 models

R4: It was corrected.

C5: L 45 not very clear- does “input data” refer to models?

R5: This sentence has been removed from the Introduction.

C6: L 54 SOM is not really a pool of C

R6: It has been corrected.

C7: L 66-71 very mixed up language. Also it’s not clear why SOM pools should follow from the objectives stated in the previous paragraph.

R7: Thank you for your comment. We understand that this is confusing following the previous paragraph. This section has been removed and this information (altered) has been moved to the Introduction.

C8: L 312-315 do the different management ‘types’ or ‘options’ refer to aspects of management that can be controlled by the models? Or are they, in the case of CCB, 5 pre-determined ‘types’ representing 5 different complete management styles?

R8: Thank you for your comment. It refers indeed to 5 different 5 pre-determined ‘types’ representing 5 different complete management styles. We have rephrased the sentence for clarification.

C9: L 318 What do you mean by ‘separate’? You mean the user can manipulate different aspects independently?

R9: That you for this comment and we understand the confusion. We meant to say that different frameworks have different simulated management options and that we should also pay attention on this. Some of them are more crucial than others for farmers, especially the ones concerning the crop yield. We have rephrased the sentence.

C10: L 324 what do you mean by ‘high value’? That they are important for stakeholders or important for SOM estimates?

R10: Thank you for the comment. We meant to say that although not directly related to the impact they have on the crop yield, they still can be valuable for farmers. We have rephrased the sentence for clarification.

C11: L 339-341 From “The same is applicable”… to “beneficial” is not needed.

R11: This sentence has been removed after the reorganization of the sections explained in the first comment.

C12: Conclusion: needs a thorough grammar review for clarity.

R12: The Conclusions section has been completely re-written, although the information is the same, we have rephrased it for clarification.

Also, since one of the reviewers required extensive editing of English language, the manuscript has been revised by a English-editing service after all the changes requested by the reviewers.

There is also a small change on the affiliation of the first author provided in line 5-6, which was incorrectly written before.

Reviewer 2 Report

In their manuscript “Identifying and comparing easily accessible frameworks for assessing soil organic matter functioning” the authors present a six frameworks that model soil organic matter dynamics with a focus on ease of use and use for practitioners.

Although the selection process and review method is not described in detail, the objective and overview presented are clearly described. A good overview to the matter of available SOM models is presented, especially to those new to the subject. I can imagine that the article is of interest to the readership of “Agronomy”. Ideally, what would have improved the paper in my eyes even more, if the authors had used one examplatory dataset for all the models to compare results point out where discrepancies in SOM simulations lie and illustrate more clearly, how the models work.

Furthermore, there are a few minor remarks that I have to the authors and that are given in detail below:

Point-by-Point comments

L21 add preposition “on” after “based”

L38 change to “as suggested earlier by (give author name- reference in brackets)”

L71 ff. Define sorts of fractions first: operational and functional fractions. I suggest you do not mix functional fractions such as polysaccharides and carbohydrates with operational fractions such as “fulvic and humic acids”

L92 change to “the mineral associated organic carbon”

Table 1 : Align cell contents in table to top (and left) in order to allow the reader to identify which content belongs to which cell –at the moment especially soil properties and management details reads as one massive text

L188 and L204 add “the” before “USA”

L218 add “the” before “UK”

Pragraph above L238 – was RothC tested successfully or not? Please give evaluating statement at the end as in other sections.

Table 2 Add info of year model was (first) developed. Check reference of CCB Model scheme (should it not be 49]?

L256 correct typo “occasional”

L261 check wording “heavy” model? Better, more precise word possible, something like “more elaborate/complicated/sophisticated” or “more multi-parametric”?

L293 change to “short life-time”

Praragraph L318-329: Very broad general content and partly double with intro. Check for redundancies.

L331 Change heading. “summary” is irritating, since it suggest you present a summary of your work here. But you introduce your scoring here so what about “Model evaluative scoring” as a more descriptive heading

Author Response

C1. In their manuscript “Identifying and comparing easily accessible frameworks for assessing soil organic matter functioning” the authors present a six frameworks that model soil organic matter dynamics with a focus on ease of use and use for practitioners.

Although the selection process and review method is not described in detail, the objective and overview presented are clearly described. A good overview to the matter of available SOM models is presented, especially to those new to the subject. I can imagine that the article is of interest to the readership of “Agronomy”. Ideally, what would have improved the paper in my eyes even more, if the authors had used one examplatory dataset for all the models to compare results point out where discrepancies in SOM simulations lie and illustrate more clearly, how the models work.

Furthermore, there are a few minor remarks that I have to the authors and that are given in detail below:

R1: Thank you so much for your kind comment. We have added a Materials and Methods section, as required for reviewer 1 and therefore the process of selection and comparison of the frameworks is better explained.

C2: L21 add preposition “on” after “based”

R2: It was corrected.

C3: L38 change to “as suggested earlier by (give author name- reference in brackets)”

R3: It was corrected.

C4: L71 ff. Define sorts of fractions first: operational and functional fractions. I suggest you do not mix functional fractions such as polysaccharides and carbohydrates with operational fractions such as “fulvic and humic acids”

R4: This section has been removed and part of it has moved into the Introduction. We mention only the different chemical and physical separation that originate different SOM pools estimated and then the link to the model simulation. 

C5: L92 change to “the mineral associated organic carbon”

R5: It has been corrected, although this sentence has been removed, the reference to MAOC in Introduction was altered has requested.

C6: Table 1: Align cell contents in table to top (and left) in order to allow the reader to identify which content belongs to which cell –at the moment especially soil properties and management details reads as one massive text

R6: It was corrected, and the formatting adopted as suggested.

C7: L188 and L204 add “the” before “USA”

R7: It was corrected.

C8: L218 add “the” before “UK”

R8: It was corrected.

C9: Pragraph above L238 – was RothC tested successfully or not? Please give evaluating statement at the end as in other sections.

R9: It has been added has requested.

C10: Table 2 Add info of year model was (first) developed. Check reference of CCB Model scheme (should it not be 49]?

R10: Thank you for your comment. The reference 49, which refers to the User manual of the CCB model does not show the model scheme. As so, we have added the scheme provided in reference 48, Franko et al. 2020.  We decided also not to add the year of development because that information was absent in some cases and therefore, we didn’t want to provide unbalanced information among the frameworks.

C11: L256 correct typo “occasional”

R11: It was corrected.

C12: L261 check wording “heavy” model? Better, more precise word possible, something like “more elaborate/complicated/sophisticated” or “more multi-parametric”?

R12: Thank you for your comment. Indeed, the word has been replaced because it was not adequate.

C13: L293 change to “short life-time”

R13: It was corrected.

C14: Praragraph L318-329: Very broad general content and partly double with intro. Check for redundancies.

R14: The paragraph has been reduced to not overlap with Introduction. Here, it remains the difference between frameworks simulating management options that can directly impact crop yield from others, which can influence farmers choices.

C15: L331 Change heading. “summary” is irritating, since it suggest you present a summary of your work here. But you introduce your scoring here so what about “Model evaluative scoring” as a more descriptive heading

R15: Thank you for your comment. The name was altered as suggested.

Also, since one of the reviewers required extensive editing of English language, the manuscript has been revised by a English-editing service after all the changes requested by the reviewers.

There is also a small change on the affiliation of the first author provided in line 5-6, which was incorrectly written before.

Reviewer 3 Report

Dear Authors,

Thank you for submitting your manuscript.

I have a few recommendations:

L. 38 - There should be the author's name and numerical reference.

Table 1 - DPM/RPM abbreviations must be explained.

L. 213-214 The model was calibrated with a long-term experiment using soil carbon data and then validated in 11 independent sites with very good results [10]. - What do very good results mean?

Table 2 - Model schemes need a higher resolution.

The page numbering is wrong

The manuscript is missing more than 20 references - compared to the reference list.

Author Response

C1. Dear Authors, Thank you for submitting your manuscript. I have a few recommendations: L. 38 - There should be the author's name and numerical reference.

R1: Thank you so much for your kind comment. The reference was corrected.

C2: Table 1 - DPM/RPM abbreviations must be explained.

R2: It was corrected.

C3: L. 213-214 The model was calibrated with a long-term experiment using soil carbon data and then validated in 11 independent sites with very good results [10]. - What do very good results mean?

R3: Thank you for your comment. It means that the simulated pools were very close to the measured ones. We have rephrased the sentence for clarification.

C4: Table 2 - Model schemes need a higher resolution.

R4: We have replaced the model schemes figures with others with better resolution. Unfortunately, some of them had already a bad resolution in the original document.

C5: The page numbering is wrong

R5: Unfortunately, we were unable to alter the page numbering in the Agronomy draft. We hope it can be corrected lately.

C6: The manuscript is missing more than 20 references - compared to the reference list.

R6: The reference list has been altered and checked again to remove possible problems. 

Also, since one of the reviewers required extensive editing of English language, the manuscript has been revised by a English-editing service after all the changes requested by the reviewers.

There is also a small change on the affiliation of the first author provided in line 5-6, which was incorrectly written before.